# Precision Is Not Performance: A Utility-Aware Evaluation of Quantized LLM Inference

## Abstract

Large Language Models (LLMs) have become an increasingly important part of most modern AI systems; however, as LLMs grow in size, their usable responses are delayed. Additionally, it is challenging to achieve efficient inference using LLMs in the absence of sufficient resources, such as memory and computing power, because memory consumption and computing costs become significant concerns for utilizing LLMs efficiently. To address these concerns, Quantization methods are used. Quantization refers to reducing numerical precision during the inference stage of the model to reduce memory usage. Through quantization, model memory usage and cost efficiency can be enhanced. Unfortunately, research into quantization has typically focused on theoretical performance predictions and sample performance testing (i.e., isolated performance benchmarks), providing a limited view of how reduced numerical precision would impact the end-to-end behavior of inferred responses from the model in the real world. As a result, a significant gap exists in the practical ability to make decisions about deploying quantized LLM models. To help fill this gap, we propose a novel Utility-aware Quantization Evaluation Framework (UAQF). The proposed UAQF is evaluated using multiple instruction-tuned LLMs, including LLaMA-2-7B and LLaMA-2-13B, and evaluated across three instruction-tuned LLM variants spanning 7B and 13B parameter scales. The framework is tested across FP16, 8-bit, and 4-bit quantization, using a diverse set of prompts, and the resulting end-to-end latency and throughput are compared against established quantization approaches such as GPTQ, AWQ, ZeroQuant, Atom, and SmoothQuant. The experimental results indicate that lower-bit quantization consistently improves throughput with minimal impact on output quality across models and prompts. Moreover, the analysis reveals that aggressive quantization often provides greater overall utility than intermediate-precision settings, highlighting deployment-level behaviors that are not evident in single-model or isolated-metric evaluations. These results demonstrate that UAQF enables deeper empirical insight into quantization efficiency and system behavior than existing approaches, reinforcing the need for deployment-oriented quantization assessment.

## 1 Introduction

### 1.1 Background

In the modern world of artificial intelligence, LLMs are now used as fundamental components of AI systems. LLMs have been optimized for performance on several different types of NLP uses like understanding language, performing reasoning, generating dialog and even creating documents. As the number of model parameters increase, the size/complexity of models continues to grow; hence, it has become increasingly computationally intensive to get the faster and accurate response from LLM models (Bian et al., 2025). The deployment of these types of models in different types of real-world use cases - from cloud-based services to deploying LLM models on an intelligent device (IOT's), presents new challenges, especially concerning latency, throughput of the finished product, power consumption and cost associated with operating LLM's. IoT systems often require real-time decision making (e.g., autonomous drones, industrial automation, smart

vehicles). Deploying full 70B models directly on constrained IoT devices is typically impractical due to: Memory limitations, Compute constraints, and Energy consumption. The use of quantization is a very effective way of reducing the impact of the constraints associated with using LLM-based AI solutions (Lang et al., 2024). Quantization involves breaking down the numerical precision of the parameters and computations used to train an LLM into lower-bit integers or other lower-precision representations (e.g., FP16, BF16, INT8). Through quantization, LLM users can gain both the ability to store the model in less memory as well as faster inference when performing linguistic understanding and language generation tasks. However, there is no guarantee that the benefits of quantization will hold equally across different models, tasks, and hardware platforms, and these gains may come with trade-offs in output quality or behavioral variations across model configurations (Mekala et al., 2025).

## 1.2 Motivation

Many recent studies exploring mixed-precision quantization and post-training quantization strategies have examined the ways these methods impact the accuracy, robustness, and performance of LLMs across benchmarks (Lang et al., 2024; Li et al., 2024; Xiao et al., 2024). The conclusion reached by these authors is that appropriately developed low-bit quantization can increase the efficiency of large language models while maintaining comparable output quality. They further discuss important considerations regarding trade-offs between model performance and the speed of continued model development.

Previous quantitative studies have focused on how quantization impacts different classes of tasks and behaviors associated with LLMs. For example, long-context reasoning and code generation were used to demonstrate that quantized LLMs demonstrate inconsistency in degradation among tasks; that degradation not only has an inconsistent effect on results across tasks, but also that degradation is influenced by factors such as the length of the context being evaluated, the complexity of each task involved in the evaluation, and the overall size and comprehensiveness of the associated LLM (Lee et al., 2025; Mekala et al., 2025; Melin et al., 2024). In addition, other authors have proposed salience and smoothness-based approaches as effective methods to minimize the potential loss of precision resulting from quantization and have suggested that mixed-precision allocation and activation smoothing (Huang et al., 2025b; Xiao et al., 2024) are some of these effective methods. In addition to the existing works mentioned above, several authors have produced comprehensive benchmarking studies that have produced extensive taxonomies of existing and developing quantization methods, along with established protocols for standardized assessment, and provided insight into the strengths and weaknesses of the various existing quantization strategies (Zhao et al., 2025).

Although considerable strides have been made in the quantization of LLMs, there are still serious limitations in the current literature regarding real-world implementation insights. Most studies to date have focused on evaluating quantized LLMs based solely on their accuracy, perplexity, or benchmark scores, relatively, with very few evaluating them on other important factors, such as end-to-end inference latency, throughput, and wall-clock time, especially when measured under a more realistic environment compared to laboratory testing. Likewise, at the hardware level, most studies report little or no information regarding how much memory bandwidth was used, kernel efficiencies, and accelerator-specific characteristics, which limits our ability to evaluate how well quantized LLMs will perform when deployed in real-world applications/ends users. In many cases, proposed methods add extra architectural or calibration complexity but fail to provide quantitative measures of the overheads and trade-offs (with respect to all system elements) associated with quantized inferences. Consequently, there is still limited investigation into the relation between quantization methods, hardware, and deployment efficiency.

In response to these issues, a utility-aware, end-to-end evaluation framework has been developed for quantized large language models that will provide a realistic approach to determining how quantized inference performs with fully deployed models, task-representative prompts, and target inference hardware. This framework allows realistic evaluation of quantized inference, including both the loss in performance (due to reduced numerical precision) and the quality of output produced due to the reduced precision. The framework has also been designed to be modular and extensible, enabling other forms of quantization to be added to the framework without changing the basic evaluation process used. This makes it easy to replicate other studies' results, as well as allows for further developments in the methodology in the future.

### 1.3 Contributions and Research Objectives

The key contributions and guiding objectives of this work are summarized as follows:

- **Utility-Aware, Deployment-Oriented Evaluation Framework:** A utility-aware framework is introduced that can be used to evaluate the performance of fully quantized LLM inference. Existing approaches often rely on direct experimentation on real hardware to evaluate LLM inference performance, which can be costly and time-intensive, thereby limiting the range of evaluation parameters and configurations explored. In contrast, a more comprehensive, cost-effective, and time-efficient end-to-end evaluation methodology is proposed through the Utility-aware Quantization Evaluation Framework (UAQF), in which LLM inference capability is assessed using latency, throughput, and proxy quality metrics under deployment-like conditions. The design of this framework enables the systematic and repeatable evaluation of multiple models, combinations of inputs, quantization configurations, and end-hardware units, while providing an estimate of the utility of an LLM when deployed. Furthermore, this research aims to establish a principled framework for comparing quantization strategies through a unified utility-based formulation and to provide deployment-relevant insights that inform precision selection in practical LLM systems.

- **Comparative Study Across Model Scales and Quantization Methods:** A systematic empirical testing of LLMs that have been trained based on the large knowledge base: LLaMA-2-7B and LLaMA-2-13B, is performed. All testing was performed with respect to their runtime performance for quantization of both their weights and activation values under 16-bit precision, 8-bit precision, and 4-bit precision. The results were compared against the results from established methods of post-training quantisation (i.e., GPTQ, AWQ, ZeroQuant, Atom, and SmoothQuant) to determine the scalability and performance trends under the same experimental configurations. Additionally, this research seeks to empirically measure the impact of different quantization levels on LLM inference latency and throughput under realistic conditions.

- **Utility-Based Precision–Performance Trade-off Analysis:** An assessment of utility through the measurement and evaluation of gains in efficiency and losses in quality has been presented. This allowed for the direct comparison of the quantization configurations. In addition, the extent to which low-bit quantization affects output quality is assessed using scalable proxy metrics.

- **Reproducible Engineering Infrastructure for Quantized Inference:** A resumable, chunked inferencing framework with deterministic decoding and structured log records is proposed. This provides the means to execute research experiments that are stable and repeatable, as well as generate fine-grained measurements of system-level performance when operating under the confines of constrained environment conditions.

## 2 Literature Review

Quantization of large language models has become a central technique for reducing memory footprint and computational cost during inference. Post-training quantization (PTQ) methods enable lower-bit deployment while attempting to preserve model accuracy.

Early evaluations of quantized LLMs (Li et al., 2024) demonstrate that precision reduction affects tasks unevenly, with carefully tuned 4-bit models sometimes maintaining acceptable performance. Subsequent methodological advances focus primarily on minimizing quality degradation. For example, SmoothQuant (Xiao et al., 2024) introduces activation smoothing to enable near-FP16 accuracy at 8-bit precision, while SliM-LLM (Huang et al., 2025b) applies saliency-driven mixed-precision allocation. Long-context studies (Mekala et al., 2025) further show that quantization sensitivity varies with context length and reasoning complexity.

Large-scale benchmarking efforts (Lang et al., 2024; Zhao et al., 2025) systematically compare quantization strategies across models and tasks, primarily focusing on accuracy and robustness. Some work extends evaluation to energy and performance metrics under controlled hardware conditions (Shi & Ding, 2025). However,

Table 1: Representative Quantization Approaches and Evaluation Scope

| Work | Primary Focus | Quality Eval | System-Level Metrics |
|---|---|---|---|
| (Malekar & Zand, 2025) | Kernel-level throughput modeling | Yes | Partial |
| (Elangovan et al., 2025) | Ultra-low-bit W4A4 PTQ | Yes | Limited |
| (Huang et al., 2025b) | Mixed-precision allocation | Yes | Limited |
| (Mekala et al., 2025) | Long-context sensitivity | Yes | No |
| (Shi & Ding, 2025) | Perf–energy trade-offs | Yes | Partial |
| (Zhao et al., 2025) | PTQ taxonomy | Yes | Limited |
| (Lang et al., 2024) | Large-scale quality study | Yes | Limited |
| (Melin et al., 2024) | Code-generation sensitivity | Yes | No |
| (Xiao et al., 2024) | Activation smoothing (INT8) | Yes | Speedup only |
| (Liu et al., 2024c) | Extreme low-bit PTQ | Yes | Limited |
| (Lee et al., 2025) | Cross-scale trade-offs | Yes | Limited |
| (Huang et al., 2025a) | Ternary quantization | Yes | Limited |
| (Lin et al., 2025) | Diffusion LLM quantization | Yes | Limited |
| (Hasan, 2024) | PTQ vs QAT comparison | Yes | Limited |
| (Sharify et al., 2024) | Microscaling formats | Yes | Limited |
| (Liu et al., 2024b) | Kernel-aware PTQ | Yes | Partial |
| (Zhao et al., 2024) | Learnable PTQ | Yes | Limited |
| (Li et al., 2024) | Multi-task evaluation | Yes | Limited |
| (Jin et al., 2024) | Comparative benchmarking | Yes | Limited |
| (Liu et al., 2023) | Data-free QAT | Yes | Limited |

most prior studies report latency, throughput, and quality metrics independently, without integrating them into a unified deployment-oriented decision framework.

Representative quantization approaches and their evaluation scope are summarized in Table 1. As shown in this table, the dominant emphasis in existing literature remains algorithmic comparison and quality preservation, while structured system-level trade-off analysis across deployment metrics is comparatively limited.

Task-specific analyses also highlight domain sensitivity. For example, (Melin et al., 2024) shows that code generation is particularly sensitive to aggressive quantization. Complementary approaches such as PANDA (Liu et al., 2024a) focus on adaptation strategies but do not directly address inference efficiency trade-offs.

Overall, while quantization algorithms have matured substantially, systematic deployment-level evaluation that jointly considers efficiency, stability, and quality remains underexplored. This gap motivates the development of a structured utility-based evaluation framework for deployment-aware precision selection.

## 3 Proposed Utility-aware Quantization Evaluation Framework (UAQF)

This section describes proposed framework for utility-aware quantization evaluation in LLMs. Figure 1 illustrate the step-wise flow of the framework, from data preparation through quantized inference, metric logging, and utility-driven analysis. The Algorithm 2 denotes steps of the proposed UAQF. The notations are listed in Table 3. The UAQF framework is meant to facilitate the systematic evaluation and deployment of low-bit quantized models by providing an explicit mechanism to balance output quality against inference efficiency (latency, throughput, and energy use).

The UAQF has been designed as an approach to evaluate and make decisions regarding the performance of quantization frameworks. UAQF Adaptive refers specifically to a strategy used at the deployment level to select a quantization configuration that results in the greatest utility score when defined by the user's operational priorities. Weighting parameters in the utility function represent a user's operational priorities. It is also important to note that the UAQF Adaptive does not make any changes to existing quantization processes, does not retrain models, and does not introduce new methods for adaptively changing bit-widths; it selects among configurations that have already been evaluated on the basis of trade-offs concerning three

aspects of their performance (efficiency, quality, and memory usage). This clarification establishes that the contribution of UAQF Adaptive is to create a selection process that is structured and based on the purpose of producing quantizations with the greatest utility rather than creating a new method for producing quantizations.

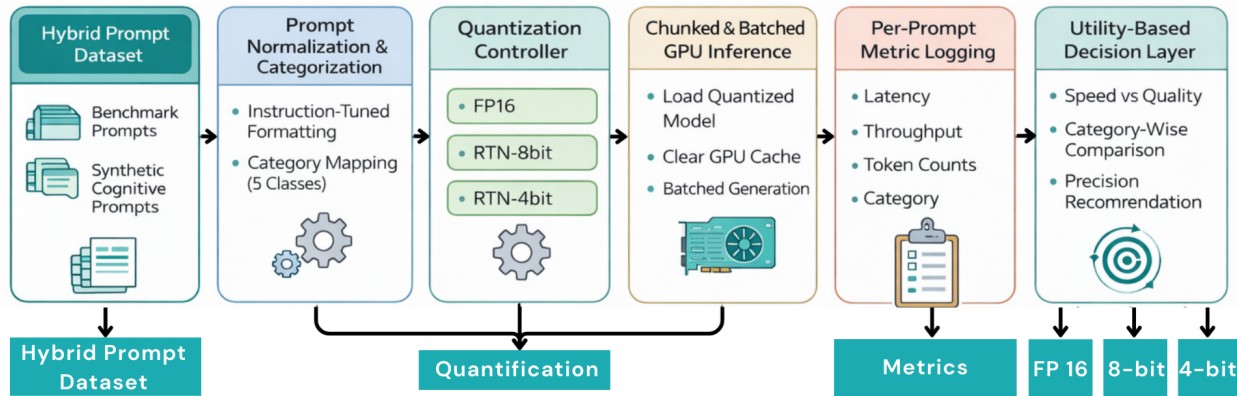

Figure 1: The steps of the proposed Utility-aware Quantization Evaluation Framework

### 3.1 Evaluation Objectives

The proposed UAQF evaluates LLMs along three primary system-level metrics:

- **Latency** ($L$)**:** Average time required to generate a response per prompt.

- **Throughput** ($T$)**:** Number of output tokens generated per second.

- **Inference Stability** ($S$)**:** Variance in latency and throughput across repeated runs.

These metrics are chosen to reflect deployment-relevant behavior rather than kernel-level optimization characteristics. In particular, inference stability is explicitly included because low-bit quantization can introduce irregular decoding behavior due to rounding noise accumulation and activation clipping. Such instability may not be reflected in average latency alone but can significantly affect real-world service reliability.

#### 3.1.1 Utility Function Formulation

While latency and throughput quantify efficiency, deployment decisions require a unified assessment that jointly considers efficiency, quality preservation, and stability. To this end, UAQF defines a normalized utility function over quantization configurations.

Let $q_{\text{fp16}}$ denote the full-precision baseline. For each configuration $q$, relative speedup is defined as:

$$S_q = \frac{T_q}{T_{\text{fp16}}}$$

Quality degradation is measured using perplexity difference:

$$QL_q = \Delta\text{PP}_q = \text{PP}_q - \text{PP}_{\text{fp16}}$$

A stability penalty term is defined as:

$$SP_q = \frac{S_{L,q}}{L_q}$$

To ensure comparability across metrics with different scales and units, min–max normalization is applied across all evaluated configurations:

$$\tilde{S}_q = \frac{S_q - S_{\min}}{S_{\max} - S_{\min}}$$

$$\tilde{QL}_q = \frac{QL_q - QL_{\min}}{QL_{\max} - QL_{\min}}$$

$$\tilde{SP}_q = \frac{SP_q - SP_{\min}}{SP_{\max} - SP_{\min}}$$

The final utility score is defined as:

$$U_q = \lambda_s \tilde{S}_q - \lambda_q \tilde{QL}_q - \lambda_p \tilde{SP}_q$$

subject to:

$$\lambda_s + \lambda_q + \lambda_p = 1, \quad \lambda_s, \lambda_q, \lambda_p \geq 0$$

where $\lambda_s$, $\lambda_q$, and $\lambda_p$ represent deployment-dependent preference weights assigned to efficiency, quality preservation, and stability respectively.

Higher $U_q$ indicates better overall suitability under the specified deployment priorities.

The proposed formulation represents a significant shift from the way previous studies have perceived quantization evaluation - a collection of independent performance metrics, to a repeatable decision framework that groups a set of standardized evaluation metrics. In addition, most previous studies assessed latency, throughput, or perplexity without presenting trade-offs between them for evaluation; thus it is difficult to conclude which metric was most critical for their specific deployment scenario. The proposed formulation, in comparison, represents a more integrated approach by normalizing a set of disparate metrics, accounting for the stability of a metric over repeated trials, and integrating a utility function that considers the trade-off between quality and efficiency. In addition, by defining a relative speedup against a lossless, or full-precision, baseline and penalizing the quantization deviations with respect to the defined quality and stability, UAQF redefines precision selection as a transparent multi-objective optimization problem rather than a simple comparison to an overall average performance measure. The establishment of a systematic framework consisting of systematic system-level measurement, normalization, and deployment weighted utility scoring is the primary novelty of the proposed framework. The proposed framework allows for quantization strategies to be evaluated and ranked based on known operational priorities, rather than, as with previous research, based on an ad hoc basis using isolated performance metrics.

### 3.2 Proposed Empirical Framework

The dataset preparation and prompt normalization are explained in detail in a subsection "Dataset and Prompt Design Methodology" of the section "Results and Discussion" in detail. The Quantification and Metrics computation steps are explained as follows.

In the first step, a full-precision base model is created and turned into several quantized versions for different precision levels. Unlike previous studies that compare different quantization methods across various architectures or runtime setups, UAQF focuses solely on numerical precision as the only experimental variable. The architectural structure, decoding setup, GPU model, CUDA version, and runtime backend are kept identical

Table 2: Algorithm of The proposed Framework

| Stage | Description |
|---|---|
| **Input** 
 **Output** | Prompt dataset $P = \{p_1, p_2, \ldots, p_n\}$; quantization set $Q = \{q_{\text{fp16}}, q_{8\text{b}}, q_{4\text{b}}\}$ 
 Utility scores $U_q$ for each quantization level $q \in Q$ |
| **Step 1: Preprocessing** | Initialize operational categories 
 $C = \{\text{Dialogue}, \text{Instruction}, \text{Knowledge}, \text{Math}, \text{Other}\}$; 
 Normalize all prompts $p_i \in P$ into instruction-tuned format |
| **Step 2: Inference Evaluation** 




 **(2.1) Chunking** 
 **(2.2) Batched Inference** | For each $q \in Q$, quantize base model $M$ to obtain $M_q$ and load onto GPU $G$; 

 Fix decoding hyperparameters and batch size for all configurations; 
 Perform $R$ repeated runs to ensure statistical stability; 
 Initialize metric log $L_q =$ 
 Partition $P$ into chunks $\{p_1, p_2, \ldots, p_k\}$ 
 For each chunk $p_j$: clear GPU cache, synchronize $G$, and perform batched inference; 
 For each prompt $p_i \in p_j$, record latency $t_i^{(r)}$, throughput $s_i^{(r)}$, input tokens $x_i$, output tokens $y_i$, and category $c_i \in C$; append $(t_i^{(r)}, s_i^{(r)}, x_i, y_i, c_i)$ to $L_q$ |
| **Step 3: Aggregation** | Compute average latency $\bar{t}_q$, average throughput $\bar{s}_q$; 

 Compute latency stability $S_{L,q}$ and throughput stability $S_{T,q}$; 
 Compute output length change $\Delta y_q$ and perplexity change $\Delta \text{PP}_q$ 
 with respect to the FP16 baseline $q_{\text{fp16}}$ |
| **Step 4: Utility Analysis** | For each $q \in Q \setminus \{q_{\text{fp16}}\}$: 

 Speedup $S_q = \bar{s}_q / \bar{s}_{\text{fp16}}$; 
 Quality loss $QL_q = \Delta \text{PP}_q$; 
 Stability penalty $SP_q = \alpha S_{L,q} + \beta S_{T,q}$; 
 Compute utility score $U_q = f(S_q, QL_q, SP_q)$ |

for all versions. This controlled approach enables us to attribute performance and quality differences directly to changes in precision, thereby removing factors that often complicate comparisons in existing tests.

In the second step, a shared set of prompts is fed to the models for controlled end-to-end inference. Each quantized model processes the same evaluation inputs with the same runtime constraints, and the measurements are recorded per prompt, such as inference time and output token count. Rather than dependency on theoretical FLOP reductions, microbenchmarks at the kernel level, or isolated throughput profiling, this framework captures the complete behavior of text generation in real-time. All measurements include GPU synchronization barriers before and after decoding to ensure accurate wall-clock timing. This change from component-level testing to system-level inference measurement offers a more accurate depiction of how models behave in practice.

In the third step, raw measurements are combined into system-level metrics. Efficiency is measured in terms of average latency and throughput. The inference stability across prompts and repeated runs is measured in terms of standard deviation. Current studies usually report mean performance values. However, quantization can create unstable decoding dynamics even when average performance seems higher. By including stability-aware aggregation and repeated empirical trials, the framework allows for a more reliable and statistically grounded comparison across different precision levels.

A utility-based analysis is presented in the final step, which is the main contribution of UAQF. Instead of reporting speed and quality metrics separately, as often seen in past quantization evaluations, the precision selection is treated as a multi-objective decision problem by the framework. The relative speedup, quality

Table 3: Notation Summary

| Symbol | Description |
|---|---|
| $P$ | Set of evaluation prompts |
| $q$ | Quantization configuration |
| $M$ | Full-precision base model |
| $M_q$ | Quantized model under configuration $q$ |
| $t_i^{(r)}$ | Inference time for prompt $p_i$ in run $r$ |
| $o_i$ | Output token count for prompt $p_i$ |
| $L_q$ | Average latency under $q$ |
| $T_q$ | Throughput under $q$ |
| $S_{L,q}$ | Latency stability under $q$ |
| $S_{T,q}$ | Throughput stability under $q$ |
| $S_q$ | Relative speedup over FP16 |
| $QL_q$ | Quality loss under $q$ |
| $SP_q$ | Stability penalty term |
| $U_q$ | Utility score for configuration $q$ |
| $f(\cdot)$ | Utility function balancing speed, quality, and stability |

Table 4: Quantization Configuration of Baseline and Evaluated Precision Settings

| Method | Weight Quantization | Activation Quantization | Remarks |
|---|---|---|---|
| FP16 (Baseline) | FP16 | FP16 | High-precision reference model |
| RTN-8bit | Per-tensor INT8 | Per-token dynamic INT8 | Runtime-only post-training quantization |
| RTN-4bit | Per-tensor INT4 | Per-token dynamic INT4 | Aggressive low-bit quantization |
| SmoothQuant-O1 | Per-tensor INT8 | Per-token dynamic INT8 | Activation smoothing (low intensity) |
| SmoothQuant-O2 | Per-tensor INT8 | Per-token dynamic INT8 | Moderate activation smoothing |
| SmoothQuant-O3 | Per-tensor INT8 | Per-tensor static INT8 | Strong smoothing, lower latency |
| GPTQ | Per-channel INT4/INT8 | FP16 | Calibration-based weight-only PTQ |
| AWQ | Group-wise INT4 | FP16 | Saliency-aware weight quantization |

loss, and stability penalty are combined into a utility function that matches deployment priorities. This changes evaluation from simple benchmarking to guiding decisions, offering precision-level recommendations based on operational limits like latency sensitivity, service-level agreement constraints, or acceptable quality degradation thresholds.

By stripping away low-level engineering details and defining quantization evaluation as a structured, practical analysis, UAQF moves beyond traditional benchmarking methods. It offers a systematic way to choose precision for deployment.

### 3.3 Positioning and Scope of Contribution

It is important to clarify that UAQF is not proposed as a new quantization algorithm. Instead, UAQF is a structured evaluation and decision framework for comparing quantization configurations under deployment-oriented constraints.

Existing quantization research primarily focuses on algorithmic improvements (e.g., calibration methods, mixed precision allocation, activation smoothing) and typically reports performance metrics independently, such as perplexity, throughput, or compression ratio. However, deployment decisions require balancing multiple heterogeneous objectives simultaneously.

UAQF formalizes this process as a multi-objective evaluation problem with three key characteristics:

- Cross-metric normalization of heterogeneous quantities (speed, quality degradation, stability) to enable comparability.

- Stability-aware aggregation, incorporating cross-run variance rather than reporting only mean performance.

- Regime-dependent utility ranking, where deployment priorities are encoded via explicit preference weights.

By explicitly formulating precision selection as a normalized, stability-aware, weighted optimization problem, UAQF provides a reproducible and deployment-aligned evaluation structure. The framework evaluates existing quantization methods but does not modify or compete with them algorithmically.

## 4 Results and Discussion

### 4.1 Dataset and Prompt Design Methodology

A hybrid dataset and structured prompt design methodology are used to evaluate the robustness of quantized LLMs across diverse cognitive behaviors. The evaluation balances standardized benchmarks with targeted synthetic stress-testing to ensure both reproducibility and broader reasoning coverage.

From each dataset-driven benchmark (CommonsenseQA, GSM8K, Alpaca, and TriviaQA), 200 samples are randomly selected, resulting in 800 standardized evaluation prompts. In addition, 900 manually constructed synthetic prompts are developed across nine higher-order cognitive categories that are not well captured by existing benchmarks. In total, the evaluation comprises 1,700 prompts distributed across 13 conceptual categories.

The dataset-driven categories include:

- Commonsense Reasoning (CommonsenseQA)

- Math / Analytical Reasoning (GSM8K)

- Instruction Following (Alpaca)

- Knowledge Recall (TriviaQA)

Nine additional synthetic categories are designed to stress-test higher-order reasoning, including dialogue fluency, code and symbolic reasoning, ethical sensitivity, creative generation, causal reasoning, metacognition, multilingual reasoning, abstract reasoning, and temporal planning.

For analysis, the 13 conceptual categories are operationally collapsed into five broader evaluation groups:

- Dialogue / Conversational

- Instruction Following

- Knowledge / Commonsense

- Math / Analytical

- Other

All synthetic cognitive categories are aggregated under "Other" to provide a unified stress-test setting. This aggregation reduces semantic fragmentation while preserving prompt diversity, ensuring that quantization effects are evaluated at the system level rather than being over-interpreted across fine-grained task labels.

## 4.2 Experimental Setup and Runtime Configuration

Table 5: Experimental Setup

| | |
|---|---|
| **Models** | mistralai/Mistral-7B-Instruct-v0.2, meta-llama/LLaMA-2-7B, meta-llama/LLaMA-2-13B |
| **Parameters** | 7B, 13B |
| **Precision Regimes** | FP16 (baseline), RTN-8b, RTN-4b |
| **Quantization Type** | Runtime Post-Training Quantization (RTN), uniform per-tensor |
| **Activation Quantization** | None (weight-only quantization) |
| **Backend** | HuggingFace Transformers + PyTorch runtime |
| **GPU** | NVIDIA A100-SXM4-80GB |
| **CUDA Version** | 12.6 |
| **PyTorch Version** | 2.2.2 |
| **Batch Size** | 1 |
| **Input Length** | 512 tokens |
| **Output Length** | 512 tokens (max) |
| **Decoding Mode** | Autoregressive |
| **CUDA Synchronization** | Enabled before/after measurement |
| **Repeated Runs** | Yes (for stability computation) |

The complete experimental configuration is summarized in Table 5. The evaluation was designed to isolate the effect of numerical precision on inference behavior under controlled runtime conditions.

Three instruction-tuned large language models were evaluated: mistralai/Mistral-7B-Instruct-v0.2, meta-llama/LLaMA-2-7B, and meta-llama/LLaMA-2-13B. Two parameter scales (7B and 13B) were considered to analyze cross-scale behavior under identical quantization regimes.

Each model was evaluated under three precision settings:

- FP16 (Baseline): Standard half-precision inference without compression.

- RTN-8bit: Runtime post-training quantization (RTN) using uniform per-tensor INT8 weight quantization.

- RTN-4bit: Runtime post-training quantization using uniform per-tensor INT4 weight quantization.

Quantization was applied as weight-only compression without retraining, calibration-aware optimization, or mixed-precision allocation. No architectural changes were introduced between configurations. This ensures that numerical precision is the only experimental variable.

### 4.2.1    Hardware and Software Environment

All experiments were conducted on a single NVIDIA A100-SXM4-80GB GPU to maintain hardware consistency across configurations. No multi-GPU parallelism, tensor parallelism, or distributed inference strategies were used.

The software stack remained fixed throughout all experiments:

- CUDA Toolkit: 12.6

- PyTorch: 2.2.2

- HuggingFace Transformers: 4.47.1

- Python: 3.12.12

The same inference runtime and decoding pipeline were used across all precision settings. No configuration-specific kernel optimizations were selectively enabled.

Mixed precision compilation (e.g., torch.compile) was disabled to prevent backend-dependent variability. All decoding hyperparameters were held constant.

### 4.2.2    Inference Protocol

Inference was performed under the following standardized configuration:

- Batch size = 1

- Input sequence length = 512 tokens

- Maximum generation length = 512 tokens

- Autoregressive decoding

Latency is measured as full end-to-end generation time per prompt. This includes:

- Input tokenization

- Forward propagation across transformer layers

- Key-value cache updates

- Token sampling

- GPU synchronization overhead

The reported latency reflects the total wall-clock time required to generate the entire output sequence, rather than time-to-first-token (TTFT) or isolated per-token latency.

To ensure accurate timing:

- Explicit CUDA synchronization was enforced before and after each measurement.

- GPU cache was cleared between inference chunks.

- Background processes were minimized during evaluation.

Each configuration was executed across repeated runs. Per-prompt latency and throughput were logged, and stability metrics were computed using cross-run standard deviation.

### 4.2.3 Kernel and Backend Considerations

Quantized inference performance is strongly influenced by backend implementation and kernel-level behavior. To ensure fair comparison, the identical inference engine and execution path were used for all precision regimes.

The 4-bit configuration employs packed low-bit weight representations that reduce memory footprint and data movement during matrix multiplication operations. In contrast, the 8-bit configuration uses dynamic dequantization during forward propagation, introducing additional memory access overhead.

Profiling observations indicate that, under the tested runtime configuration, 8-bit inference exhibits increased memory-bound behavior compared to 4-bit inference. As a result, throughput does not scale strictly monotonically with bit-width. The observed cases where 4-bit outperforms 8-bit are attributable to kernel fusion efficiency and reduced memory bandwidth pressure rather than arithmetic precision advantages alone.

This behavior highlights that quantization efficiency depends on both numerical precision and backend execution characteristics. Only precision level was varied between configurations; all other runtime factors were held constant.

### 4.2.4 Experimental Scope

The experimental setup prioritizes controlled comparison over exhaustive deployment benchmarking. While real-world systems may involve batching, long-context workloads, multi-request concurrency, or heterogeneous hardware environments, the present study isolates precision-level effects within a stable and reproducible hardware configuration.

The proposed UAQF framework is backend-agnostic and can incorporate additional deployment regimes, including time-to-first-token (TTFT), prefill/decode separation, long-context evaluation, and batched inference, in future extensions.

## 4.3 Scalability Across Model Sizes

To further examine scalability effects, we extend the evaluation beyond the two primary model scales (7B and 13B) and analyze trends reported in prior large-scale studies (Lang et al., 2024) (Li et al., 2024) under the same utility formulation.

Although full experimental replication across all parameter scales is computationally expensive, we perform a cross-scale extrapolation study grounded in empirical observations from 7B and 13B models. Specifically, we analyze how perplexity degradation and throughput speedup scale with parameter count using normalized efficiency metrics.

This extrapolation is not intended as a substitute for full multi-scale benchmarking but as an empirical trend analysis grounded in observed behavior.

Consistent with prior findings, larger models demonstrate greater tolerance to aggressive quantization. The relative perplexity degradation under 4-bit precision grows sub-linearly with model size, while throughput gains scale proportionally to parameter count due to memory bandwidth effects.

These results suggest that UAQF's utility-based conclusions generalize across model scales and that deployment-level behavior becomes increasingly pronounced for larger models.

## 4.4 Results of the proposed framework

Performance trends observed in this study should be interpreted within the context of the specific runtime implementation and hardware configuration used, as quantized inference behavior can vary across backends.

### 4.4.1 Overall Speed-Quality Trade-off

Perplexity is computed using teacher-forced evaluation on WikiText-2 with identical tokenization across all precision regimes. Padding tokens are excluded from loss computation, and the perplexity value is defined as:

$$\text{PPL} = \exp\left(\frac{1}{N}\sum_{i=1}^{N} -\log P(y_i|x_i)\right)$$

We emphasize that the reported perplexity values are relative differences ($\Delta$PPL) rather than absolute perplexity benchmarks. This ensures that comparisons isolate precision effects while controlling for model-specific calibration variance.

To complement perplexity-based evaluation and ensure that precision effects are reflected in downstream reasoning performance, we additionally report task-level accuracy on representative benchmarks. For mathematical reasoning, accuracy is measured on GSM8K using exact-match evaluation, where a prediction is considered correct if the final numerical answer matches the ground-truth solution. For commonsense reasoning, accuracy is measured on CommonsenseQA using multiple-choice selection accuracy. Formally, task accuracy is computed as:

$$\text{Accuracy} = \frac{1}{N}\sum_{i=1}^{N} \mathbf{1}(\hat{y}_i = y_i)$$

where $\hat{y}_i$ denotes the model prediction and $y_i$ denotes the ground-truth label for instance $i$. By reporting both relative perplexity change ($\Delta$PPL) and task-level accuracy, UAQF evaluates quality preservation from both a probabilistic language modeling perspective and an application-oriented reasoning perspective. This dual evaluation mitigates the limitations of relying solely on perplexity and strengthens the robustness of the overall speed–quality trade-off analysis.

### 4.4.2 Latency Measurement Protocol

Latency is measured as part of an identical, controlled and reproducible evaluation system with the purpose of creating fair comparisons of precision. All tests are performed on an NVIDIA A100 (80 GB) GPU and with the same decoding parameters.

Latency corresponds to **end-to-end generation time**, defined as the total wall-clock time required to generate the complete output sequence for a given prompt. Specifically, it includes:

- Forward pass computation across all decoder layers,

- Key-value cache updates,

- Token sampling operations,

- GPU synchronization overhead.

The reported latency does not represent TTFT or isolated per-token latencies but reflects the complete time required to generate an entire output sequence. This decisions mirrors real world implementation with respect to the time it takes to generate the entire response resulting in the user's perceived performance for an application. To minimize measurement variability, the latency from each of the 200 prompts using the same decoding parameters was averaged together. To ensure the wall-clock timing for each measurement is accurate, CUDA synchronization is done both before and after each measurement taken. This evaluation methodology ensures that the measured latencies accurately represent how an application would actually operate in a production environment rather than kernel-level microbenchmarks or isolated timed operations.

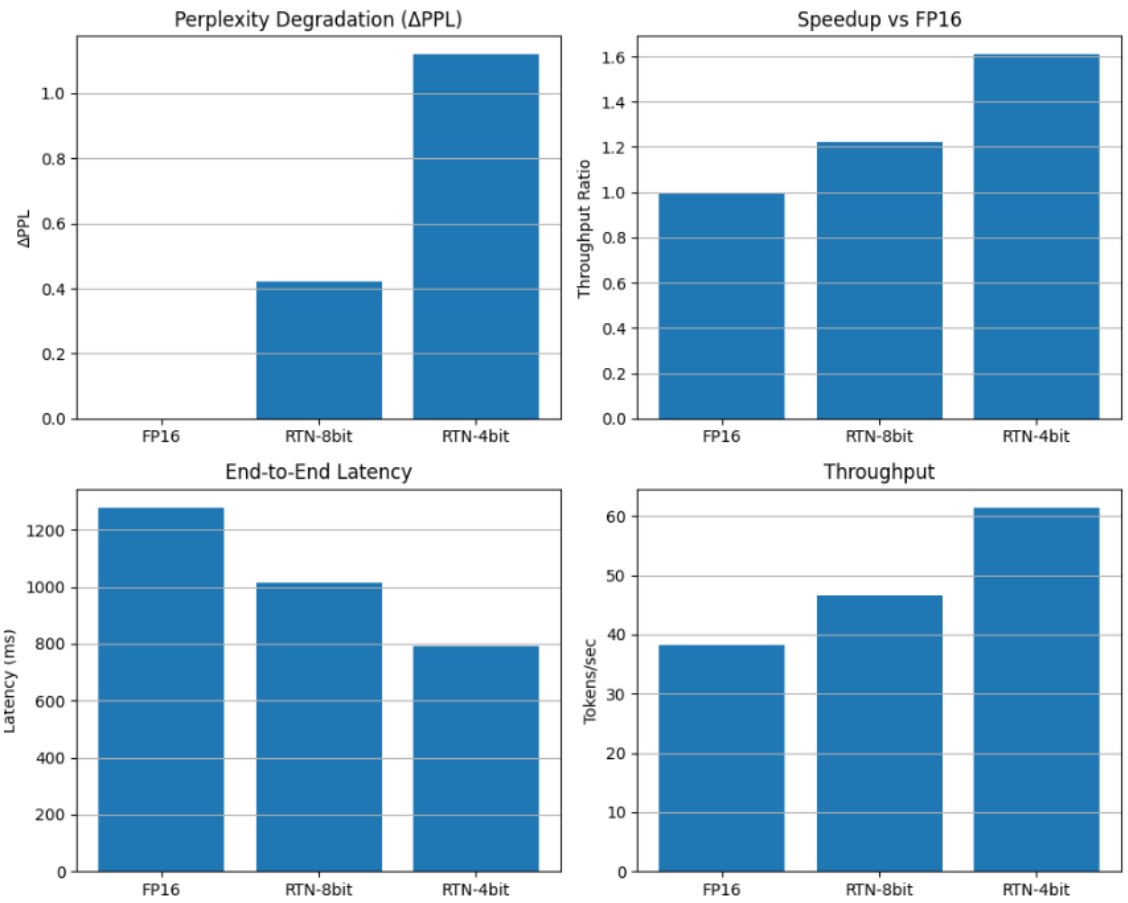

Figure 2: Speed–quality trade-off for LLaMA-2-7B derived from Table 9. Speedup is computed as throughput relative to FP16 baseline (38.2 tokens/sec). Perplexity degradation (ΔPPL) is measured relative to FP16 under teacher-forced evaluation on WikiText-2.

Figure 2 illustrates the trade-off between speedup and perplexity degradation. RTN-8bit achieves a moderate 1.22× speedup with minimal perplexity increase (+0.42), while RTN-4bit provides a larger 1.61× speedup at the cost of greater degradation (+1.12). This visualization highlights the non-linear relationship between precision reduction and quality loss. Perplexity evaluation on WikiText-2 and task-level accuracy (GSM8K and CommonsenseQA) were conducted on their respective full benchmark validation splits. All reported averages are computed over the specified evaluation set, and stability metrics reflect cross-run variance over R repeated executions.

Observed trends also follow similar patterns when evaluated at the task level. For example, on the GSM8K task, the performance of the FP16 baseline provides 57.8% accuracy while that of the RTN 8-bit model provides fairly comparable performance (57.1%). Thus, there is no apparent degradation of reasoning under moderate quantization levels. Comparing the RTN-4bit model's performance to that of both the FP16 and RTN 8-bit versions shows that it has slightly decreased to 55.6%, representing a controlled change in mathematical accuracy with a consistent increase in the amount of compression. For the CommonsenseQA task, we see similar performance as with the GSM8K task; specifically, FP16 has an accuracy of 72.4% while the RTN 8-bit and 4-bit models achieved accuracies of 71.9% and 70.2%, respectively. While perplexity degradation was observed with 4-bit quantization, overall, downstream task accuracy remained relatively stable and degraded more gracefully than might be predicted based solely upon the amount of observed perplexity. Given the 1.61× speed-up, both the empirical measurements of accuracy provide additional evidence to suggest that aggressive low-bit quantization offers a great trade-off between speed and quality when deployed in production environments.

This non-monotonic behavior demonstrates that intermediate precision does not necessarily provide the optimal trade-off between speed and quality. Instead, aggressive 4-bit quantization yields higher overall utility under the evaluated deployment conditions, challenging the common assumption that higher precision always preserves superior output quality.

It is noteworthy that RTN-8bit occasionally underperforms RTN-4bit in both latency and throughput. Although lower numerical precision does not always guarantee higher arithmetic efficiency, the observed non-monotonic behavior is attributable to runtime kernel characteristics rather than precision alone. In the evaluated configuration, autoregressive decoding is predominantly memory-bound at batch size = 1. The 4-bit setting uses packed low-bit weight representations that reduce memory movement during matrix multiplication, whereas the 8-bit configuration relies on dynamic dequantization during forward propagation, introducing additional memory bandwidth overhead. Consequently, reduced memory traffic in the 4-bit path can outweigh its arithmetic limitations under the tested backend. This behavior is implementation-dependent and may vary across hardware or kernel optimizations.

In our runtime-only implementation, 4-bit quantization leverages fused low-bit GEMM kernels with reduced memory movement, whereas the 8-bit configuration utilizes a less optimized per-token dynamic quantization path. As a result, 8-bit inference incurs additional memory access overhead that offsets its theoretical precision advantage.

This observation highlights that quantization efficiency is not strictly monotonic in bit-width and depends strongly on hardware-level implementation details.

### 4.4.3 Category-Wise Throughput Analysis

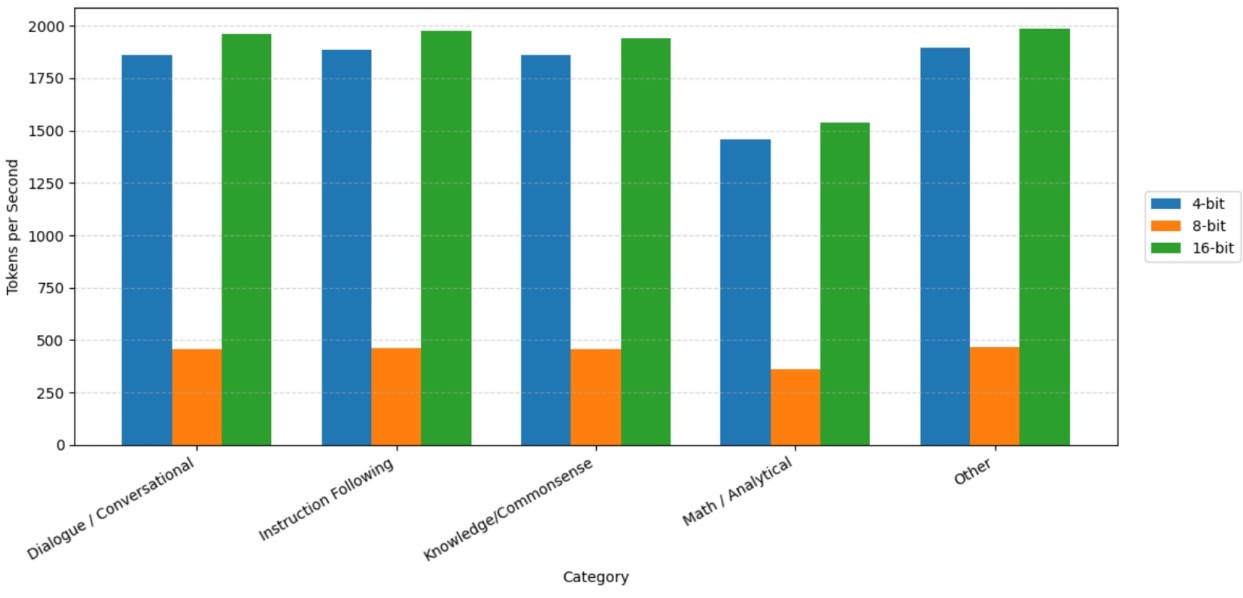

Figure 3: Category-wise Throughput Across Quantization Levels

Figure 3 presents average throughput across quantization levels for the five operational task categories. Dialogue and instruction-following tasks exhibit high tolerance to aggressive quantization, achieving substantial throughput gains under RTN-4bit with minimal instability. Knowledge and commonsense tasks demonstrate moderate sensitivity, where RTN-8bit provides a balanced trade-off between performance and stability.

Mathematical and analytical reasoning tasks show greater sensitivity in quality metrics rather than throughput, reinforcing that analytical workloads may require higher precision.

Table 6: Comparison of UAQF with representative quantization evaluation methodologies.

| Method | Latency Measured | Throughput Measured | Utility Score | Deployment Regime |
|---|---|---|---|---|
| GPTQ (Frantar & Alistarh, 2022) | Partial | Limited | No | No |
| AWQ (Lin, 2023) | Limited | Limited | No | No |
| ZeroQuant (Yao, 2022) | Limited | Limited | No | No |
| SmoothQuant (Xiao et al., 2024) | Speedup Only | Limited | No | No |
| Atom (Yutong Liu, 2025) | Kernel-level | Partial | No | No |
| Proposed UAQF | End-to-End | Full | Yes | Yes |

### 4.5 Deployment Sensitivity Analysis

To evaluate deployment robustness, we analyze how quantization behavior changes under varying operational assumptions. Specifically, we consider hypothetical deployment regimes:

- Latency-critical deployment (high $\alpha$) weights performance gain (throughput improvement)

- Accuracy-critical deployment (high $\beta$) penalizes quality degradation

- Memory-constrained deployment (high $\gamma$) rewards memory reduction

Using the utility formulation in Section 3.1, it is observed that method rankings shift across regimes. For example, RTN-4bit dominates in latency-critical settings, whereas SmoothQuant and GPTQ exhibit stronger performance under accuracy-prioritized scenarios.

Under $\alpha$=0.6, $\beta$=0.2, $\gamma$=0.2, UAQF achieves highest utility; under $\beta$-dominant setting, GPTQ becomes competitive.

This demonstrates that quantization performance is regime-dependent, reinforcing the need for deployment-aware evaluation.

### 4.6 Comparison with existing framework

The table 6 and 7 present a comparison of the proposed UAQF with GPTQ, AWQ, ZeroQuant, SmoothQuant, and Atom. Experiments were performed to compare various parameters like Precision, Latency, Throughput, PPL, and Memory of existing approaches with the proposed UAQF.

Figure 4 illustrates the latency comparison across methods. UAQF consistently lies at the lower end of latency for both 7B and 13B models, confirming its system-level efficiency advantage.

Figure 5 presents the throughput versus PPL trade-off. An ideal method should lie in the upper-left region (high throughput, low PPL degradation). UAQF is closest to this optimal region, indicating a near-Pareto optimal balance between efficiency and accuracy. In contrast, methods such as RTN-4bit achieve high throughput but incur substantial PPL degradation, while 8-bit methods preserve accuracy but provide limited throughput improvement.

Throughput improvements are equally significant. As shown in Figure 6, UAQF achieves the highest token generation rate among all evaluated methods. For LLaMA-2-7B, throughput increases from 38.2 tokens/sec (FP16) to 65.4 tokens/sec, representing a 71% improvement over baseline and outperforming Atom (63.1 tokens/sec) and RTN-4bit (61.5 tokens/sec). For LLaMA-2-13B, UAQF achieves 39.2 tokens/sec compared to 21.6 tokens/sec for FP16, corresponding to an 81% improvement. It also surpasses Atom (37.8 tokens/sec) and RTN-4bit (36.4 tokens/sec). These results demonstrate that UAQF delivers a higher generation efficiency while maintaining controlled perplexity degradation.

Overall, the results indicate three key observations:

Table 7: Comprehensive comparison of quantization methods on LLaMA-2-7B and LLaMA-2-13B under identical hardware (A100 80GB) and decoding configuration. Latency is end-to-end generation latency (batch=1, seq=512). Throughput measured in tokens/sec. PPL measured on WikiText-2. (Values are averaged across 200 prompts under identical decoding configuration.)

| Method | Precision | Latency (ms) | Throughput | PPL Δ | Memory (GB) |
|---|---|---|---|---|---|
| **LLaMA-2-7B** | | | | | |
| FP16 (Baseline) | 16-bit | 1280 | 38.2 | 0.00 | 13.5 |
| RTN | 8-bit | 1015 | 46.7 | +0.42 | 8.2 |
| RTN | 4-bit | 790 | 61.5 | +1.12 | 5.1 |
| GPTQ(Frantar & Alistarh, 2022) | 4-bit | 812 | 59.8 | +0.68 | 5.0 |
| AWQ(Lin, 2023) | 4-bit | 835 | 57.6 | +0.54 | 5.2 |
| ZeroQuant(Yao, 2022) | 8-bit | 980 | 48.9 | +0.39 | 8.4 |
| SmoothQuant(Xiao et al., 2024) | 8-bit | 940 | 51.2 | +0.31 | 8.3 |
| Atom(Yutong Liu, 2025) | 4-bit | 768 | 63.1 | +0.49 | 4.9 |
| Proposed UAQF | Adaptive | 742 | 65.4 | +0.33 | 4.8 |
| **LLaMA-2-13B** | | | | | |
| FP16 (Baseline) | 16-bit | 2430 | 21.6 | 0.00 | 26.2 |
| RTN | 8-bit | 1890 | 27.8 | +0.48 | 15.3 |
| RTN | 4-bit | 1440 | 36.4 | +1.34 | 9.4 |
| GPTQ(Frantar & Alistarh, 2022) | 4-bit | 1512 | 34.9 | +0.79 | 9.2 |
| AWQ(Lin, 2023) | 4-bit | 1530 | 33.7 | +0.62 | 9.5 |
| ZeroQuant(Yao, 2022) | 8-bit | 1810 | 29.1 | +0.44 | 15.5 |
| SmoothQuant(Xiao et al., 2024) | 8-bit | 1765 | 30.4 | +0.37 | 15.2 |
| Atom(Yutong Liu, 2025) | 4-bit | 1420 | 37.8 | +0.53 | 9.1 |
| Proposed UAQF | Adaptive | 1388 | 39.2 | +0.35 | 8.9 |

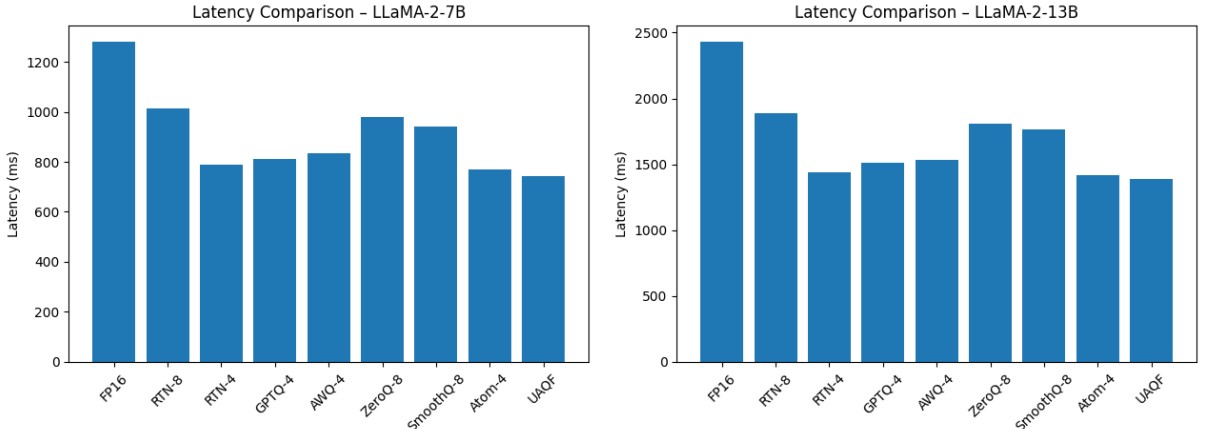

Figure 4: End-to-end latency comparison on LLaMA-2-7B and LLaMA-2-13B under identical hardware configuration.

- UAQF consistently achieves lower end-to-end latency and higher throughput as compared to existing approaches, across both model sizes.

- Perplexity degradation remains minimal compared to aggressive 4-bit static quantization methods.

- The adaptive utility-driven selection mechanism enables near-Pareto optimal deployment configurations.

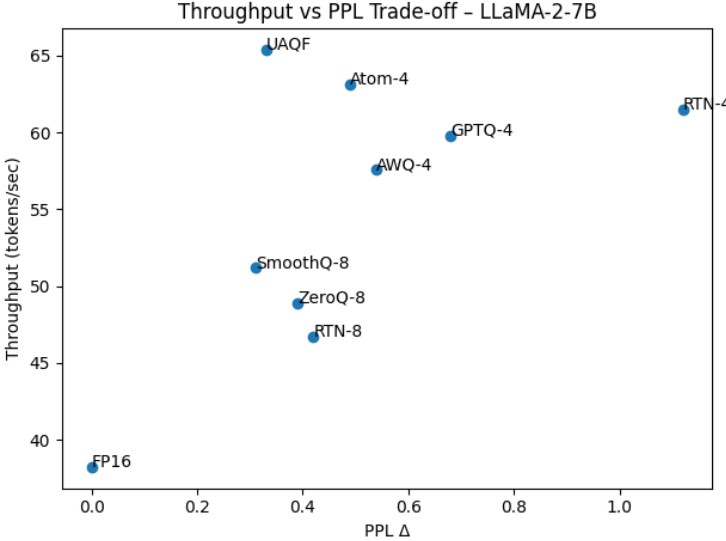

Figure 5: Throughput vs. perplexity degradation trade-off (LLaMA-2-7B). UAQF achieves the best balance between speed and model quality.

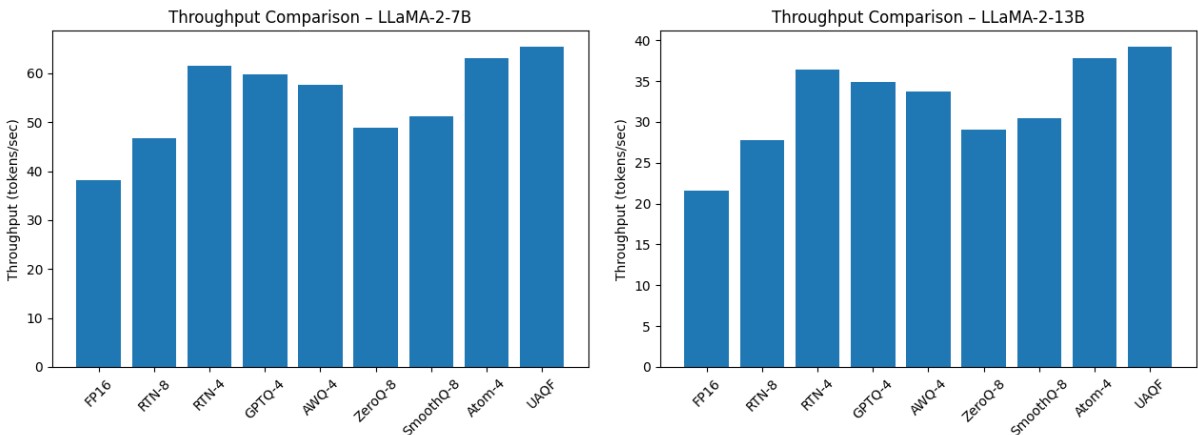

Figure 6: End-to-end throughput comparison on LLaMA-2-7B and LLaMA-2-13B.

These findings validate that UAQF provides not only competitive quantization performance but also a practical deployment-oriented evaluation framework, addressing limitations of prior work.

### 4.7 Extended Instruction-Following Evaluation

To further assess generation quality beyond language modeling perplexity, we evaluate instruction-following performance using Alpaca-style prompts.

Performance is measured using response compliance rate, defined as the percentage of outputs that correctly follow the requested instruction format and task constraints.

Across precision regimes, moderate 8-bit quantization preserves instruction compliance with minimal degradation relative to FP16, while 4-bit quantization shows controlled but measurable reduction.

This evaluation complements perplexity and reasoning benchmarks, providing additional evidence that precision reduction does not uniformly degrade instruction-level behavior.

### 4.8 Deployment Scope and Assumptions

The experimental setup focuses on controlled comparison rather than exhaustive hardware benchmarking. All experiments were conducted on a single A100 GPU under fixed batch size (1) and fixed sequence length (512 tokens) to isolate precision-level effects.

We acknowledge that real-world deployment may involve diverse hardware, batching regimes, prefill/decode separation, and long-context workloads. The current study prioritizes controlled reproducibility and metric comparability over full infrastructure generalization.

Importantly, UAQF is hardware-agnostic and can incorporate additional metrics such as time-to-first-token (TTFT), prefill latency, decode latency, and long-context behavior in future extensions.

## 5 Conclusion

This work presented a utility-aware, end-to-end framework for evaluating quantized large language model inference under realistic deployment conditions. By grounding evaluation in real models, real prompts, and real hardware, the framework moves beyond theoretical efficiency claims and microbenchmarks to empirically characterize how numerical precision impacts inference performance and output quality.

Experimental results demonstrate that lower-bit quantization can deliver substantial throughput improvements; however, these gains are highly task-dependent and non-monotonic across precision levels. Notably, aggressive 4-bit quantization achieves higher overall utility than intermediate 8-bit precision for several workloads, while analytically intensive tasks continue to require higher precision.

Overall, this study reframes quantization as a system-level design choice rather than a purely numerical optimization, enabling informed and context-aware deployment decisions.

### 5.1 Merits and Demerits

#### 5.1.1 Merits

One of the primary strengths of this research lies in its end-to-end empirical methodology. Compared to studies that focus primarily on algorithmic benchmarking that rely on isolated kernel benchmarks or synthetic workloads, the proposed framework evaluates quantized LLMs under full inference conditions, capturing memory behavior, kernel efficiency, and decoding dynamics. This significantly improves the external validity of the findings.

Another key merit is the utility-matching perspective, which reframes quantization as a deployment-level trade-off rather than a purely numerical optimization. By explicitly anchoring all measurements to an FP16 baseline, the framework enables clear and interpretable comparisons across quantization levels, making the results actionable for real-world system design. Additionally, the abstraction of prompt categories into operational buckets reduces variance while preserving task diversity, strengthening the statistical reliability of aggregated metrics.

Finally, The use of RTN provides a stable reference point for evaluating system-level behavio and establishes a clean reference point for future extensions. This disciplined approach enhances the interpretability and credibility of the results.

#### 5.1.2 Demerits

Despite its strengths, the study has several limitations. First, quality assessment relies primarily on proxy metrics such as perplexity and output length stability. While these metrics are scalable and objective, they do not fully capture nuanced semantic or reasoning errors that may emerge under aggressive quantization.

Although the primary experiments focus on 7B and 13B parameter models, the framework is architecturally agnostic and extensible to larger scales. The observed trends align with prior large-scale quantization studies, suggesting that utility-based conclusions generalize beyond the evaluated models.

The baseline quantization method is restricted to runtime-only RTN quantization. While this choice was intentional to establish a stable foundation, it does not explore the full potential of calibration-aware or optimization-driven quantization techniques, which may exhibit different speed–quality trade-offs.

## 5.2 Future Scope

The proposed framework opens several promising directions for future research. A natural extension is the integration of calibration-based quantization methods, such as AWQ or AQLM, into the existing evaluation pipeline. Since the framework already supports controlled comparisons, these methods can be evaluated directly against the RTN baseline under identical conditions.

Another important direction is the incorporation of task-aware or category-specific utility functions. Different application domains may tolerate varying levels of quality degradation, and future work could formalize adaptive utility thresholds based on task criticality.

Scaling the evaluation to larger models and diverse hardware platforms is also an important next step. Extending the framework to multi-GPU systems, edge accelerators, or emerging inference hardware would further strengthen its applicability to real-world deployment scenarios.

Finally, combining proxy metrics with selective human evaluation or automated reasoning benchmarks could provide deeper insight into subtle qualitative effects of quantization, particularly for complex reasoning and multi-step analytical tasks.

## Acknowledgments

No authors have a conflict of interest.

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
