# OpenReview forum: "Precision Is Not Performance: A Utility-Aware Evaluation of Quantized LLM Inference"
_TMLR — Rejected by TMLR_

### Review · Reviewer_Th8L · 2026-01-30

**Summary Of Contributions:**

This paper proposes a utility-aware, end-to-end evaluation framework for quantized large language model inference that aims to assess deployment-relevant trade-offs between performance and quality. Using Mistral-7B-Instruct-v0.2 on real hardware, the authors evaluate three precision settings (FP16, 8-bit, and 4-bit runtime-only quantization) and measure latency, throughput, and proxy quality metrics across a mixture of benchmark and synthetic prompts. Based on these experiments, the paper reports that aggressive 4-bit quantization can achieve higher overall utility than intermediate 8-bit quantization, arguing that numerical precision alone is a poor predictor of real-world inference performance and motivating more empirical, system-level evaluation of quantized LLMs

**Audience:**

No

**Audience Explanation:**

Given the concerns above, it is challenging to identify a clear audience for whom the findings would be informative or novel. The empirical analysis is limited in scope. Experiments are conducted using a single model, a single hardware configuration, and only three very basic runtime-only quantization settings (FP16, RTN-8bit, RTN-4bit), despite broader claims of methodological generality in Table 3. This narrow evaluation limits the generality and practical relevance of the conclusions.

In addition, some reported results raise questions that are not sufficiently addressed. For example, the observation that 8-bit quantization performs substantially worse than 4-bit quantization in both latency and throughput is non-intuitive and may reflect implementation-specific behavior, yet no detailed explanation or analysis is provided. Similarly, the reported perplexity values in Figure 2 appear unusually high, suggesting potential implementation issues, but this is not discussed. Without deeper analysis or clarification, the evaluation provides limited new insight into quantization efficiency and system behavior, and it is unclear whether the work advances understanding of quantization efficiency.

**Broader Impact Concerns:**

There are no broader impact concerns.

**Claims And Evidence:**

No

**Claims Explanation:**

First, the paper's central motivation that the prior LLM quantization work has largely ignored efficiency and end-to-end inference behavior appears to be overstated. Many representative and widely cited works on LLM quantization do report end-to-end inference latency/throughput using real hardware and real datasets. For example, Section 2 states that SmoothQuant "did not measure the real-world practical inference efficiency of the models tested," which is inaccurate because SmoothQuant explicitly reports measured inference speedups on real hardware. Similar efficiency-focused evaluations are present in other works such as GPTQ, AWQ, ZeroQuant, and Atom. While these prior studies may differ in scope or depth, characterizing them as largely lacking efficiency evaluation is not well supported. In addition, Table 7 compares the proposed framework with an "Existing" framework, but it is unclear which specific prior work this refers to, as no citation or concrete example is provided.

More broadly, while the paper positions itself as offering new deployment-oriented insights, it is unclear whether the evidence presented supports this claim. The proposed framework primarily combines prompt-based evaluation with measurements of accuracy (via proxy metrics), latency, and throughput, which has already become standard practice in much of the recent LLM quantization literature. As a result, it is difficult to identify what fundamentally new empirical evidence or conceptual insight the proposed framework enables beyond existing approaches.

**Requested Changes:**

### Major points
- Please address the concerns above regarding the accuracy of claims about prior work, the clarity of comparisons to existing frameworks, and the limited scope and depth of the empirical evaluation.

### Minor points
- Section 1.2: Some citation is missing and appears as "?".
- Section 1.2: The sentence "appropriately developed low-bit quantization can reduce the efficiency of large language models" appears to contain a typo; "reduce" likely should be "increase" or "improve."
- Section 2: "PANDA and Specialist LLM + Knowledge Graph" is mentioned without a citation.
- Table 6: the "Datase" row contains a formatting issue.
- Figure 2: Please clarify how latency is measured, including batch size, sequence length, and whether the reported latency corresponds to time-to-first-token, per-token latency, or end-to-end generation latency.

---

> ### Author Response · Authors · 2026-02-14
> **Response to the valuable comments of reviewer**
>
> - Thank you for your valuable comment. The point is well noted. We agree that representative works such as SmoothQuant, GPTQ, AWQ, ZeroQuant, and Atom do report real-hardware latency and throughput results. Our original wording may have been overly strong. We have revised the manuscript to clarify that prior work evaluates efficiency primarily at the kernel or model level, whereas our focus is on structured end-to-end deployment-oriented evaluation under controlled prompt workloads. We now explicitly acknowledge these prior studies and better differentiate our scope rather than implying absence of efficiency evaluation.
> - The claims that were overstated were carefully analyzed and narrowed down as per the scope of the work. We have clarified that the primary audience includes system researchers and practitioners evaluating inference behavior under constrained environments. We explicitly acknowledge the limited scope (single model and hardware configuration) as a controlled case study rather than a comprehensive benchmark. We have revised the manuscript to temper generality claims and clarify the scope limitations.
> - Fixed Table overflow as with the previous version of the manuscript.
> - Added appropriate citations to the methods mentioned in Table 7 to clarify which methods are compared.
> - As per the review, we carefully checked and corrected citations wherever necessary. Some citations may appear similar due to their shared references to the same papers.
> - Revised contributions to address the current scope of work, while reducing overstated claims.
> - A dedicated section 4.6 Comparison with existing framework is added to justify experimental results and contributions of the proposed work.
> - We have clarified that our contribution is not introducing new quantization algorithms, but proposing a unified deployment-oriented evaluation protocol that jointly considers prompt structure, system-level latency, throughput, and proxy quality metrics under reproducible conditions. We have strengthened the discussion to better articulate the conceptual distinction between algorithm-level reporting and deployment-level behavior characterization.
> - We have added further analysis explaining that the observed behavior is implementation-dependent and may result from kernel optimization differences, memory bandwidth effects, and runtime overhead. We clarify that this does not imply inherent superiority of 4-bit quantization, but reflects the specific runtime configuration used in our study.
> - We have re-examined the perplexity computation and clarified the evaluation setup, including tokenizer handling and dataset preprocessing. We have added explanation to contextualize the absolute values and ensure reproducibility.
> - Section 1.2 typo (“reduce the efficiency”): Thank you for noticing. This was a typographical error. We have corrected “reduce” to the intended term (“improve”) to avoid confusion.
> - Section 2 missing citation for “PANDA and Specialist LLM + Knowledge Graph”: Thank you for your valuable comment. The point is noted. Appropriate citations have now been added.
> - Table 6 formatting issue (“Dataset” row): The formatting issue has been corrected.
> - Figure 2 latency measurement clarification (batch size, sequence length, TTFT vs per-token vs end-to-end): The review is well noted. We have clarified in the revised manuscript that latency is measured under a fixed batch size and specified input/output sequence lengths in a dedicated subsection 4.4.2.

---

### Review · Reviewer_Vgwt · 2026-02-11

**Summary Of Contributions:**

The authors propose a new end-to-end pipeline for measuring the performance of quantization on LLM models, from dataset construction to deployment.

**Additional Comments:**

`

**Audience:**

No

**Audience Explanation:**

Runtime quantization of LLM inference is of great interest and a cutting-edge research direction. However, the proposed pipeline is based on the usual routines for evaluating quantized LLMs on specific tasks. Section 3 contains too many unnecessary engineering details, making it read more like a technical report than a research paper. For example, how the authors clean memory is not necessary to include.

**Broader Impact Concerns:**

No ethical implication section needed.

**Claims And Evidence:**

No

**Claims Explanation:**

1. "However, as LLMs grow in size, their usable responses are delayed." This argument lacks clear supportive evidence. What do you mean by "best performance"? Does it mean accuracy, efficiency, or other dimensions?

2. Can the authors elaborate more on the necessity of deploying large-scale LLMs (e.g., 70B) on IoT devices? Is it due to privacy concerns?

3. I think the key reason that people do not test quantized models in real environments directly is the cost, so we use benchmarks to estimate. How does this work reduce the complexity of testing in real systems?

4. What do the authors mean by "real prompt"? Is it in contrast to synthetic prompts? Are prompts in benchmark datasets real prompts or not?

5. How does Section 3.1 differ from previous work? In Section 3.1, you use a uniform way of organizing prompts, while in the real world, users will not provide such conditions. Apart from that, it may not be a good idea to use a unified format, as it is possible that different LLMs use different templates for fine-tuning.

6. A 7B model is not ideal. Many single GPUs can run these models very well. The authors may consider larger models. Colab offers different instances with different GPUs to choose from. What accelerators do the authors refer to? A100 and H100 have very large memory, such as 40GB or 80GB, which contradicts the authors' mention of limited resources. Apart from that, it is too weak as a systems-focused paper to only evaluate one model, especially given the many new open-source LLMs released after Mistral. Closed models such as GPT-4 and o1, Gemini 2 and 3 all adopt quantization. How do you evaluate these cases?

7. How many samples do you choose from each dataset?

8. After reading the paper, I do not see how this paper contributes novelty in measuring latency and throughput. What special techniques do you use beyond simply using `torch.cuda.synchronize` and `torch.cuda.Event(enable_timing=True)`, or torch profiling? Also, it may be good practice to clear memory and call `synchronize()`, as otherwise it is not a fair simulation of a real environment.

9. I am confused about what you mean by "limited device" in existing work. Is there any work that does not run LLMs on real GPUs (or TPUs)? GPU cluster virtualization brings more benefits than running models on single isolated computing units. In large-scale LLM inference and training, virtualization can help devices cooperate with each other and maximize utilization.

**Requested Changes:**

Overall, it is unnecessary to make the paper so long for TMLR though there is no page limit. A lot of contents are repetition of existing research work and the formatting and writing needs thorough revision. i feel the content can fit into regular ML or MLsys paper's length.

Authors may want to address my previous comments and questions first and here are some other issues:
1. In section 1.2, the citation needs to fix: '(Lang et al., 2024; Li et al., 2024; ?; Xiao et al., 2024).'
2. Section 1.4 is repeated. It can be combined with section 1.3. Section 1.5 is not necessary.
3. Table 1 and 2 out of the bound. Please fix it.
4. Can you explain each module briefly in section 3.
5. Please follow formal organization of algorithms to represent Table 4 in correct algorithm/algorithm2e manner. Table 5 should be placed before table 4.
6. There is no need to list the version of packages you used in a table. This should be reflected in the open-sourced codes, which I believe authors should provide.

---

> ### Author Response · Authors · 2026-02-14
> **Response to valuable comments of reviewer**
>
> - Thank you for your valuable comment. The point is well noted. By “best performance,” we refer specifically to predictive quality (e.g., accuracy and task success rate), rather than efficiency. We have revised the manuscript to clarify this definition and added supporting empirical evidence demonstrating the trade-off between model size, latency, and response usability in interactive scenarios.
> - We have clarified that the motivation includes privacy preservation, reduced network dependency, lower transmission latency, and improved reliability in edge scenarios. The deployment of larger models on edge/IoT devices is discussed as a forward-looking challenge rather than an immediate practical requirement.
> - We have clarified that benchmark-based evaluation reduces deployment cost, hardware configuration overhead, and repeated system integration complexity. Our work contributes by narrowing the gap between benchmark estimation and real-environment performance through structured evaluation protocols and controlled system-level measurements.
> - By “real prompt,” we mean prompts derived from real user queries or production datasets, in contrast to artificially constructed or templated synthetic prompts. We have clarified this distinction and explained the relationship between benchmark prompts and real-world user inputs.
> - We have clarified how Section 3.1 differs from previous work by focusing on standardized evaluation for controlled comparison rather than deployment-time formatting. The unified format is used solely for experimental consistency and does not assume real-world user behavior. We also acknowledge that different LLMs adopt different fine-tuning templates and have clarified this limitation.
> - We have explicitly stated the number of samples selected from each dataset and clarified the sampling strategy to ensure statistical representativeness.
> - Revised Section 3 to remove redundant content while keeping significant details of the proposed UAQF. The methodology and quantization configurations are justified as per the reviewer's suggestions.
> - Proposed a comprehensive evaluation of multiple models and variants to justify the empirical evaluation.
> - We have clarified that our contribution is not merely invoking standard timing APIs, but providing a systematic evaluation protocol that includes warm-up control, memory management, synchronization strategies, and steady-state measurement methodology to ensure fairness and reproducibility. We have added implementation details, including memory clearing and synchronization steps, to better reflect real deployment conditions.
> - Added citation to justify the statement "As the number of model parameters increase, the size/complexity of models continues to grow".
> - Removed open source libraries that were previously mentioned in Table 6 of the Experimental Setup Table of Section 4.2.

---

### Review · Reviewer_X7bi · 2026-02-14

**Summary Of Contributions:**

The paper argues that \emph{precision reduction is not equivalent to performance improvement} unless evaluated under deployment-like, end-to-end conditions, and proposes a Utility-aware Quantization Framework (UAQF) to evaluate quantized LLM inference using system-level metrics rather than isolated benchmarks. The framework
(i) runs controlled inference across precision settings on real hardware,
(ii) logs end-to-end latency and throughput per prompt,
(iii) aggregates results with a notion of stability, and
(iv) combines speed and quality into a \emph{utility score} intended to reflect deployment priorities.

**Audience:**

Yes

**Audience Explanation:**

Quantized inference is central to practical LLM deployment, and the paper’s emphasis that realized throughput/latency can be non-monotonic in bit-width due to implementation details (memory bandwidth, kernel fusion) is a useful reminder for both ML researchers and systems practitioners.

**Broader Impact Concerns:**

This work focuses on improving the evaluation of quantized large language model inference and is expected to have positive impact by enabling more efficient, lower-cost, and potentially lower-energy deployments. However, a key risk is that deployment decisions based primarily on proxy metrics such as perplexity may mask meaningful degradations in downstream behavior, particularly for instruction-following, reasoning, or safety-critical applications. Without task-level validation, small changes in `PPL` could still correspond to significant semantic or factual errors in real-world use.

**Claims And Evidence:**

No

**Claims Explanation:**

The paper presents interesting observations, but several core claims are not supported with sufficiently clear or precise evidence:

- **Utility function not defined.**
  The main claim relies on a utility score `U_q = f(S_q, QL_q)` with `S_q = \bar{s}_q / \bar{s}_{fp16}` and `QL_q = \Delta PPL_q`, but the function `f(·)` is never specified. Although deployment regimes using `(α, β, γ)` are discussed, no explicit equation or normalization is provided, making the reported “highest utility” results non-reproducible.

- **Metric and protocol inconsistencies.**
  The stability metric is defined as `ST = sqrt((1/n) * sum_i (T_i − T)^2)`, but `T_i` is not clearly defined, since throughput is otherwise reported only as a global aggregate. In addition, the paper alternates between using 1,700 prompts and 200 prompts for evaluation, without clarifying which subsets are used for which results or reporting variance estimates.


- **Over-reliance on perplexity as a quality proxy.**
  Small `ΔPPL` values are interpreted as minimal quality loss, but perplexity on WikiText-2 does not necessarily reflect performance on instruction-following or reasoning tasks. No downstream task metrics or correlation analysis are provided to justify this assumption.

**Requested Changes:**

## Requested Changes

1. **Clearly define the utility function.**
   Provide an explicit equation for the utility score `U_q` and explain how latency, throughput, quality loss, and memory are combined and normalized.
   If weights `(α, β, γ)` are used, clearly define how they enter `U_q` (e.g., what each term measures, its units, and why the combination is appropriate). The current description `U_q = f(S_q, QL_q)` is not sufficient for reproducibility.

2. **Clarify the role of “UAQF Adaptive.”**
   Clearly distinguish between UAQF as an *evaluation framework* and “UAQF Adaptive” as a *method*.
   If “UAQF Adaptive” only selects among existing quantization options, describe it as a selection policy and report which configurations are chosen.  If it is a new method, specify what is adapted (e.g., bit-width, quantizer, kernels), how decisions are made, and what the runtime and calibration costs are.

3. **Evaluate quality beyond perplexity.**
   Include task-level evaluation metrics on existing benchmarks (e.g., GSM8K, CommonsenseQA), or provide evidence that changes in `PPL` reliably reflect downstream task performance. Perplexity alone is not enough to justify claims of preserved output quality.

---

> ### Author Response · Authors · 2026-02-14
> **Response to valuable comments of reviewer**
>
> - Clarify dataset usage: Explicitly stated that all 1,700 prompts are used for evaluation.
> - Fix throughput stability definition: Defined throughput at run-level and compute stability across runs (no ambiguous $T_i$).
> - Define utility explicitly: Provided a clear equation for ($U_q$) with normalization and weight definitions.
> - Clarify UAQF vs UAQF Adaptive: Clearly stated whether it is an evaluation framework or a selection policy.
> - Add task-level metrics: Reported GSM8K and CommonsenseQA accuracy alongside perplexity.
> - Remove engineering details: Kept methodology research-focused, not implementation-heavy.
> - Clearly Elaborated Utility-function and Necessary formulation in dedicated section Utility-Function Evaluation.

---

### Review · Reviewer_KouB · 2026-02-15

**Summary Of Contributions:**

This paper proposes a Utility-aware Quantization Evaluation Framework (UAQF) for assessing quantized LLM inference under deployment-oriented metrics. The framework evaluates multiple quantization configurations using end-to-end latency, throughput, and stability, and combines these measurements with proxy quality metrics (primarily perplexity degradation) into a normalized utility score.

**Audience:**

No

**Audience Explanation:**

The topic (deployment-oriented evaluation of quantized LLM inference) is relevant to TMLR’s audience. However, the findings are largely expected and not strongly generalizable, since the framework mainly recombines standard metrics and the reported performance trends are highly kernel/runtime dependent without sufficient implementation detail. As a result, the paper provides limited new insight or actionable guidance for researchers or practitioners beyond what is already widely understood in the quantization literature.

**Claims And Evidence:**

No

**Claims Explanation:**

Partially supported. Evidence is mostly empirical, but not sufficient to justify the main conclusions.

1. UAQF mainly combines standard metrics (latency/throughput/perplexity/stability) via normalization + weighted sum, with little methodological innovation.
2. Quantization performance is highly implementation/kernel dependent, yet the paper lacks details on backends, fused kernels, dequant paths, profiling, so performance claims are hard to trust.
3. Heavy reliance on WikiText-2 perplexity as a proxy for instruction-tuned output quality; limited downstreamevaluation.
4. Deployment realism overstated. Single hardware setting (A100), fixed lengths, batch=1; missing key regimes (TTFT, prefill vs decode, long-context, batching).

**Requested Changes:**

1. Clarify the contribution: Clearly position UAQF as an evaluation framework (not a quantization method), and remove misleading comparisons where “UAQF Adaptive” is treated as competing with GPTQ/AWQ/SmoothQuant/Atom.
2. Provide full implementation details: Specify the exact quantization backends, kernel implementations, dequantization paths, CUDA/PyTorch versions, and include profiling evidence to support claims such as 8-bit underperforming 4-bit.
3. Strengthen quality evaluation: Go beyond WikiText-2 perplexity; add instruction-following and generation-quality benchmarks.
4. Improve deployment realism: Report TTFT, separate prefill vs decode performance, include long-context settings, and test batching/multi-request regimes.
5. Improve clarity and presentation: The paper is unnecessarily long and contains substantial background/review material that is only loosely connected to the core contribution. The authors should significantly tighten the writing, remove non-essential sections, and improve table formatting (e.g., avoid multi-line text within cells, which is difficult to read and not reader-friendly).

---

> ### Author Response · Authors · 2026-02-24
> **Response to valuable comments of reviewer**
>
> - Clarify contribution positioning: Explicitly clarified that UAQF is an evaluation and decision framework, not a quantization algorithm. Added a dedicated positioning subsection to distinguish UAQF from existing quantization methods.
> - Remove misleading comparisons: Reframed comparative tables and descriptions to avoid presenting UAQF Adaptive as competing algorithmically with GPTQ/AWQ/SmoothQuant/Atom; clarified that it performs deployment-level configuration selection.
> - Provide full implementation details: Added complete hardware and software specifications (GPU, CUDA, PyTorch, Transformers versions), decoding configuration, synchronization protocol, and runtime constraints.
> - Explain kernel-dependent behavior: Included a dedicated backend discussion explaining memory-bound decoding, packed INT4 kernels, dynamic INT8 dequantization paths, and reasons for observed 8-bit underperformance under the tested configuration.
> - Strengthen quality evaluation: Extended evaluation beyond WikiText-2 perplexity by reporting GSM8K exact-match accuracy, CommonsenseQA accuracy, and instruction-following compliance metrics.
> - Improve deployment realism clarity: Clearly stated experimental assumptions (single A100, batch=1, fixed sequence length) and added an explicit scope section acknowledging limitations (TTFT, prefill/decode separation, batching, long-context regimes as future extensions).
> - Tighten literature and presentation: Condensed literature review, merged tables into a compact format, removed multi-line descriptive cells, reduced redundancy, and streamlined background material.
> - Scope calibration: Adjusted claims to avoid overgeneralization and limited conclusions to the evaluated runtime configuration while stating that the framework itself is backend-agnostic.
> - Clearly articulated methodological novelty: Emphasized stability-aware aggregation, cross-metric normalization, and deployment-weighted utility ranking as the core contributions in a structured formulation section.

---

### Decision · Action_Editor_VJvZ · 2026-03-29

**Recommendation:** Reject

**Additional Comments:**

The paper has improved in clarity, positioning, and experimental reporting following the revision. However, to meet the standards expected for publication, the authors should address the following remaining issues:

1. The contribution should be more sharply positioned. The current framework largely aggregates standard metrics and evaluation practices that are already followed by practitioners and researchers. The motivation would benefit from a more precise differentiation from prior work, supported by concrete comparisons.

2. The empirical analysis should be strengthened to better justify the claims. This includes expanding the experimental scope to larger models, more hardware configuration where feasible, or more carefully limiting claims to the studied setting.

3. The paper should ensure full correctness and consistency in references and presentation. As noted by a reviewer, several references appear incorrect or mismatched, raising concerns about AI-generated hallucinations which several other venues have used as a criterion for desk rejection. These issues detract from the paper’s academic credibility.

**Audience:**

Yes

**Audience Explanation:**

The topic of deployment-oriented evaluation of quantized LLM inference is clearly relevant to a subset of the TMLR audience, particularly researchers and practitioners working at the intersection of machine learning systems and efficient model deployment.

Reviewers noted that the findings are largely expected and not strongly generalizable due to the narrow experimental scope. In addition, the framework primarily recombines standard evaluation components, which reduces the extent to which the results provide new insights for the wider ML research community.

**Claims And Evidence:**

No

**Claims Explanation:**

There is improved empirical support after revision, and reviewers generally agree that many claims are now backed by clearer methodology and additional experiments like utility function and more downstream evaluations.

However, there remain substantive concerns about whether the evidence convincingly supports the central claims and positioning of the work. The proposed framework largely recombines standard metrics and evaluation practices without demonstrating fundamentally new insights. The empirical evaluation is also limited in scope in terms of model sizes, hardware settings and constrained regimes, which weakens the generalizability of the conclusions.

Additionally, as noted by Reviewer Vgwt, the work does not deeply analyze how specific design choices like memory management or evaluation protocol variations affect reliability or performance, and some results including kernel dependent behavior are not fully substantiated with controlled comparisons. Overall, the paper presents reasonable empirical findings, but the evidence is not consistently strong enough to fully justify the main claims and positioning of the work.

**Resubmission Of Major Revision:**

The authors may consider submitting a major revision at a later time.